# Spontaneous body wall contractions stabilize the fluid microenvironment that shapes host–microbe associations

**Janna C Nawroth[1,2,3†], Christoph Giez[4†], Alexander Klimovich[4], Eva Kanso[1]\*, Thomas CG Bosch[4]\***

[1]Aerospace and Mechanical Engineering, University of Southern California, Los Angeles, United States; [2]Helmholtz Pioneer Campus and Institute of Biological and Medical Imaging (IBMI), Helmholtz Munich (GmbH), Neuherberg, Germany; [3]Chair of Biological Imaging at the Central Institute for Translational Cancer Research (TranslaTUM), School of Medicine, Technical University of Munich, Munich, Germany; [4]Zoological Institute, Kiel University, Kiel, Germany

**Abstract** The freshwater polyp *Hydra* is a popular biological model system; however, we still do not understand one of its most salient behaviors, the generation of spontaneous body wall contractions. Here, by applying experimental fluid dynamics analysis and mathematical modeling, we provide functional evidence that spontaneous contractions of body walls enhance the transport of chemical compounds from and to the tissue surface where symbiotic bacteria reside. Experimentally, a reduction in the frequency of spontaneous body wall contractions is associated with a changed composition of the colonizing microbiota. Together, our findings suggest that spontaneous body wall contractions create an important fluid transport mechanism that (1) may shape and stabilize specific host–microbe associations and (2) create fluid microhabitats that may modulate the spatial distribution of the colonizing microbes. This mechanism may be more broadly applicable to animal–microbe interactions since research has shown that rhythmic spontaneous contractions in the gastrointestinal tracts are essential for maintaining normal microbiota.

**\*For correspondence:**
kanso@usc.edu (EK);
tbosch@zoologie.uni-kiel.de
(TCGB)

[†]These authors contributed equally to this work

**Competing interest:** The authors declare that no competing interests exist.

## Editor's evaluation

This important work studies the spontaneous contractions (SC) of the *Hydra* body wall and presents a mathematical model of nutrient transport to hypothesize the role of SC on maintaining the microbiota. The solid evidence presented yields insights into the functional implications of the SC and the increased nutrient update obtained from mixing the local fluid environment through body wall contractions. The main result represents an important observation about the role of hydrodynamics on organism behavior and its relation to diffusive chemical transport processes.

## Introduction

The millimeter-scale freshwater polyp *Hydra* with its simple tube-like body structure (**Figure 1A**) and stereotypic movement patterns is a popular biological model organism for immunology (**Bosch and McFall-Ngai, 2021**; **Bosch, 2014**; **Bosch, 2013**; **Klimovich and Bosch, 2018a**; **Schröder and Bosch, 2016**), developmental and evolutionary biology, and neurobiology (**Klimovich and Bosch, 2018a**; **Bosch et al., 2017**). Despite *Hydra*'s relevance for fundamental research, one of the animal's most salient behaviors remains a mystery since its first description in 1744 (**Bayer et al., 1744**): it is unclear why *Hydra* undergoes recurrent full-body contraction-relaxation cycles, approximately one every

**Figure 1.** The freshwater polyp *Hydra*, a model system for the role of spontaneous body wall contractions in shaping microbe biogeography. (**A**) Polyp colonized with fluorescent labelled betaproteobacteria. (**B**) A representative contractile activity pattern of an individual polyp recorded over 8 hr (left). Each dash on the timeline represents an individual spontaneous contraction–relaxation cycle (right). (**C**) Schematic representation of a *Hydra* with a fluid boundary layer surrounding the polyp with the glycocalyx layer adjacent to the polyp's tissue. Inset: tissue architecture covered outside by the mucus-like glycocalyx that provides the habitat for a specific bacterial community on the interface with the fluid boundary layer. (**D**) Dense community of fluorescently labeled bacteria (main colonizer *Curvibacter* sp.) colonizing the polyp's head. (**E**) The biogeography of *Hydra*'s symbionts under undisturbed/control conditions follows a distinct spatial colonization pattern along the body column (n = 4). (**F**) Bacteria mostly colonizing the foot region include *Pseudomonas*, *Flavobacterium*, and *Acidovorax* (n = 4). Scale bar: 500 μm.

10–30 min (*Figure 1B*), while attached to the substratum. Most, if not all, animals exhibit such spontaneous contractions of muscular organs and body walls to pump internal fluids, create mixing flows, feed, locomote, or clean surfaces (*Schierwater et al., 1991*; *Kremien et al., 2013*; *Trojanowski et al., 2016*; *Rich, 2018*; *Kornder et al., 2022*). Other animals perform rhythmic pulsations, which differ from spontaneous contractions by occurring in predictable intervals and usually serve similar functions. For example, rhythmic pulsation of tentacles in corals, first noted by Lamarck nearly 200 years ago, lead to increased turbulent mixing, which enhances photosynthesis via fast removal of excess oxygen and prevents refiltration of surrounding water by neighboring polyps (*Kremien et al., 2013*). However, *Hydra*'s size, morphology, and spontaneous contraction characteristics do not fit any of the known functions of rhythmic or spontaneous contractions. In particular, *Hydra*'s small dimensions likely

prevent it from the generation of turbulent mixing (*Purcell, 1977*). Here, we sought to investigate the functional underpinnings of *Hydra*'s spontaneous contractions from a fluid mechanics perspective. We hypothesized that the spontaneous contractions might generate fluid transport patterns of relevance to *Hydra*'s microbial partners that colonize the so-called glycocalyx on the outer body wall (*Schröder and Bosch, 2016*; *Fraune et al., 2015*; *Figure 1C*). Typically, the composition and distribution of symbiotic microbial communities residing at solid–liquid interfaces, such as in the gut and lung, are regulated both by interaction with host cells and by the properties of the fluid medium (*Fang et al., 2021*; *Franzenburg et al., 2013*). Therefore, we asked whether *Hydra*'s microbiome might be influenced by fluid flow associated with spontaneous contractions.

## Results

First, we mapped the abundance and distribution of microbiota that reside on the glycocalyx (*Figure 1C and D*). The glycocalyx is a multilayered extracellular cuticle covering *Hydra*'s ectodermal epithelial cells, has mucus-like properties, and shapes the microbiome by providing food and antimicrobial peptides (*Franzenburg et al., 2013*). Regional differences in the glycocalyx have not been reported; however, using 16S rRNA profiling on different sections of *Hydra*, we found that different microbial species favor different regions along the body column, including foot- and head-dominating microbiota (*Figure 1E and F*). Based on these intriguing data and recent insights into how fluid flow shapes spatial distributions of bacteria (*Wheeler et al., 2019*), we hypothesized that the spontaneous contractions might generate fluidic microhabitats that facilitate the microbial biogeography along the body column.

### Kinematics and flow physics of individual spontaneous contractions

To explore this hypothesis, we used video microscopy and analyzed the animal's body kinematics to assess the differential impact of spontaneous contractions on the fluid microenvironment. We examined the contraction activity (*Figure 2A*) of multiple animals over 8 hr and recorded an average contraction frequency of 2.5 spontaneous contractions per hour (*Figure 2—figure supplements 1 and 2*, *Figure 2—figure supplement 3* control conditions), corresponding to an average duration ($T_{IC}$) of the inter-contraction intervals of 24 min. To arrive at estimates rooted in probability theory that take into consideration the distribution of $T_{IC}$, we collected the measured $T_{IC}$ from all animals in the form of a histogram (*Figure 2—figure supplement 3B*, control condition). The resulting distribution of $T_{IC}$ is best fit by an exponential distribution, implying that contraction events of individual polyps follow a stochastic Poisson process. We computed the average frequency based on the exponential fit to these biological data. We found that the so-computed contraction frequency is equal to 2.9 contractions per hour, which is slightly larger than the 2.5 contractions per hour obtained by taking a direct average of the data.

Each stage of the inter-contraction intervals and spontaneous contraction cycle is characterized by stereotypic kinematic patterns of *Hydra*'s motion trajectories and velocities (*Figure 2A*, bottom). During inter-contraction intervals, which on average last tens of minutes (*Figure 2—figure supplement 3B*), *Hydra*'s body column remains nearly fully extended and slowly revolves at full length around its foothold, tracing a cone-shaped volume over time (*Figure 2B*, *Video 1*). By contrast, spontaneous contraction events typically last a few seconds and are characterized by a stepwise shortening of the body column with one or more peak contraction events until the body column has shortened to 20% or less of its original length (*Figure 2A–C*). A peak contraction is defined as any part of the spontaneous contraction during which *Hydra* contracts at a rate of 25% change in body length per second or more. spontaneous contractions are frequently accompanied by axial rotation in a spiraling downward motion (*Figure 2A and B*, *Video 1*) and are typically followed by slow re-extension to inter-contraction interval length (*Figure 2A*) in a random direction (*Figure 2—figure supplement 4*). We quantified the speeds and timescales that different regions of *Hydra*'s body column experience during inter-contraction intervals and spontaneous contractions. During spontaneous contractions, specifically during peak contractions characterized by simultaneous linear shortening and axial rotation, the oral region accelerates to peak velocities $U_{PC}$ of 10 mm/s. This is almost two orders of magnitude faster than maximal head speeds $U_{IC}$ during inter-contraction intervals, which are on the order of 0.1 mm/s in both longitudinal and axial directions (*Figure 2B and C*). Further, peak contraction rarely

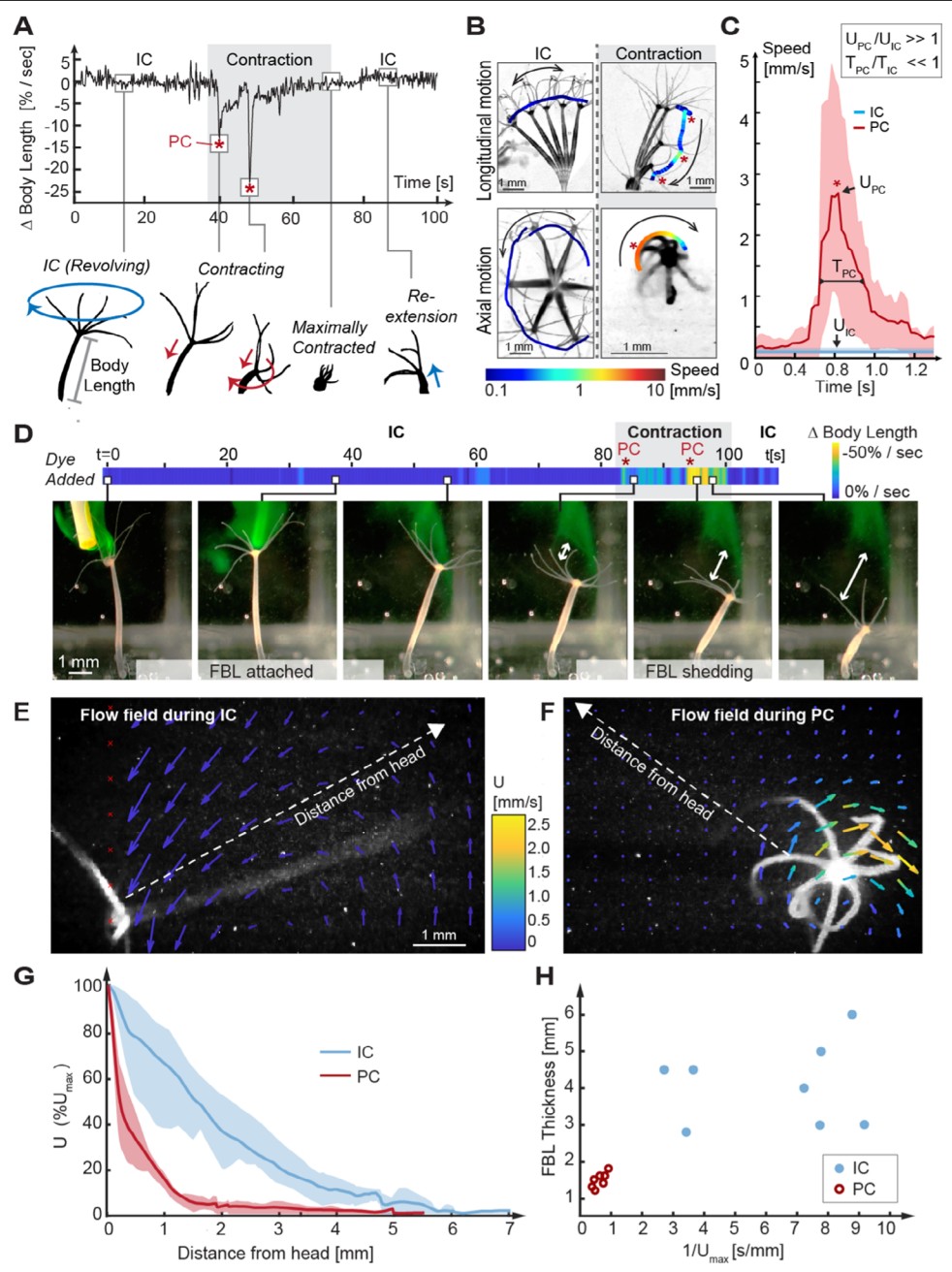

**Figure 2.** Kinematic and fluid dynamics analysis of individual contraction events reveal the shedding of the fluid boundary layer (FBL) during each spontaneous contraction event. (**A**) Top: representative plot of the change in relative body length in *Hydra* as a function of time shows transition from an inter-contraction interval (IC) to a spontaneous contraction with two peak contraction events (PC, asterisks), and the return to an IC interval. Bottom: typical kinematic pattern associated with IC intervals and contractions. Arrows indicate distinct body trajectories during IC intervals (blue) compared to contractions (red). (**B**) Typical body trajectories in longitudinal and axial plane during IC intervals and spontaneous contractions visualized by time-lapse microscopy. Maximal speeds (log10 scale) are indicated by color-coded trajectories of the head (oral end or tentacles). Trajectories are slightly offset to avoid obscuring the animal. Asterisks denote PC events. (**C**) Comparison of maximal velocities near head reached during IC intervals (n = 3 animals) and PC (n = 5 animals). Lines: average curves. Shaded areas: interquartile range. Inset: relative scaling of PC duration ($T_{PC}$), IC interval duration ($T_{IC}$), PC velocity magnitudes ($U_{PC}$), and IC interval velocity magnitudes ($U_{IC}$). (**D**) Application of a fluorescent dye reveals existence of a FBL during the IC interval and its shedding upon a contraction event. A representative time-lapse series. White arrows indicate FBL shedding after PCs, that is, the growing separation between the original, stained FBL and

*Figure 2 continued on next page*

*Figure 2 continued*

*Hydra*'s head. (**E**) Quantification of *Hydra*'s flow velocity field during IC intervals and (**F**) during a typical PC with axial rotation (top view). Flow vectors and velocities are indicated by color-coded arrows. (**G**) Relative change of fluid flow speed as a function of distance from *Hydra*'s surface measured along dotted lines in (**E**, **F**) during IC intervals (blue) and during PCs with rotation (red). Lines: average curves. Shaded areas: interquartile range. (**H**) FBL thickness, defined as distance from *Hydra* at which 90% freestream speed is reached, is inversely correlated to maximal flow speed ($U_{\mathrm{max}}$).

The online version of this article includes the following figure supplement(s) for figure 2:

**Figure supplement 1.** The experimental setup for measuring and manipulating the contraction rates in *Hydra*.

**Figure supplement 2.** Behavioral analysis of *Hydra* over 8 hr.

**Figure supplement 3.** Spontaneous contractions are modeled mathematically as a Poisson process.

**Figure supplement 4.** The 2D space covered by *Hydra*'s resting motion and re-extensions in the period of nine contractions is revealed by this overlay of detected motion.

**Figure supplement 5.** Typical flow speed profiles derived from the particle imaging velocimetry (PIV) data.

**Figure supplement 6.** Spontaneous contraction frequencies of *Hydra* when placed in small liquid volumes can reach up to 12 contractions per hour (CPH).

last longer than $T_{\mathrm{PC}} = 1\mathrm{s}$ and hence operate at 1000-fold smaller timescales than inter-contraction intervals (with mean duration of ca. 24 min, i.e., 1440 s).

To assess the effects of *Hydra*'s body kinematics on its fluid environment, we computed the Reynolds number Re = $\ell U/\mu$ at the tentacles near *Hydra*'s mouth, where the tentacle diameter is $\ell = 0.1$ mm, the kinematic viscosity of water at 20°C is $\mu = 1.0034$ mm²/s, and the tentacle speed $U$ is in mm/s. The Re number compares the effects of fluid inertia to viscous drag forces. During inter-contraction intervals, $U = U_{IC}$ is on the order of 0.1 mm/s, and Re is on the order of 0.01. At such low Re number, fluid flow is dominated by viscous effects and distinguished by the presence of a laminar and expansive fluid boundary layer. The fluid boundary layer can be envisioned as an envelope of fluid enclosing *Hydra* and moving at the same velocity as *Hydra*'s surface. The thickness of the fluid boundary layer is defined as the distance from the surface at which the fluid speed decreases by 90% compared to *Hydra*'s speed, that is, the distance at which the fluid is no longer following *Hydra*'s surface (*Vogel, 2020*). During peak contraction, *Hydra*'s speed $U = U_{\mathrm{PC}}$ is on the order of 10 mm/s, Re increases to the order of 1, which indicates greater inertial effects that result in thinning of the fluid boundary layer (*Schlichting and Gersten, 2000*).

To label the fluid boundary layer and assess the animal–fluid interactions experimentally (*Nawroth and Dabiri, 2014*), we added fluorescent dye adjacent to *Hydra*'s oral region while in the inter-contraction interval state. In one representative recording (*Figure 2D*, left, *Video 2*), the dye faithfully followed the fully extended animal's slow revolutions during the inter-contraction interval for more than 1 min, qualitatively confirming the presence of a stably attached fluid boundary layer. Intriguingly, as the animal underwent a spontaneous contraction (with two peak contraction events), the dye separated from the animal's surface and stayed behind as *Hydra* retracted, indicating shedding of the original fluid boundary layer (*Figure 2D*, right, *Video 2*). Since the animals tend to slowly

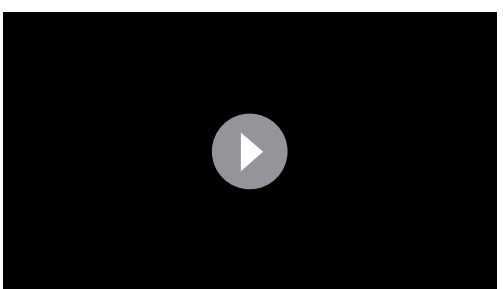

**Video 1.** *Hydra* alternating between rest (inter-contraction time) and spontaneous contractions.
https://elifesciences.org/articles/83637/figures#video1

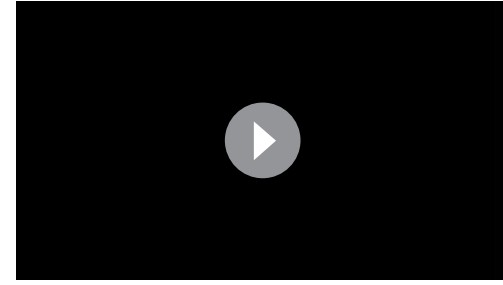

**Video 2.** Dye visualization of fluid boundary layer (FBL) dynamics during rest (inter-contraction time), when it remains stable, and during a spontaneous contraction, when much of the FBL is shed.
https://elifesciences.org/articles/83637/figures#video2

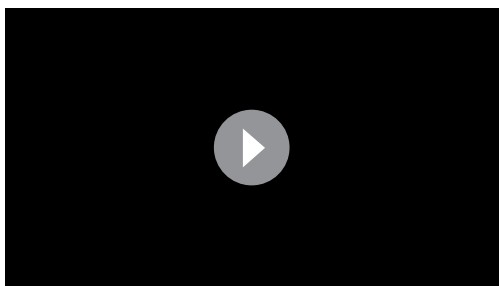

**Video 3.** Particle image velocimetry to quantify fluid boundary layer dynamics during rest (inter-contraction time) and spontaneous contractions.
https://elifesciences.org/articles/83637/figures#video3

re-extend into a random direction (*Figure 2—figure supplement 4*), their chances of re-encountering the shed fluid are small. To confirm these observations quantitatively and to measure the dynamically changing fluid boundary layer thickness, we recorded the microscale fluid motion using particle imaging velocimetry (PIV) as described previously (*Nawroth et al., 2017*). During inter-contraction intervals, fluid parcels at distances far from the body column followed the animal's trajectory, reflecting an expansive fluid boundary layer of almost one full-body length in thickness (fluid boundary layer thickness = 5 mm) (*Figure 2E and G*, *Figure 2—figure supplement 5A*, *Video 3*). In contrast, during peak contractions, only the fluid close to *Hydra*'s surface accel-

erated with the body, whereas fluid at further distances remained unaffected (fluid boundary layer thickness = 1.5 mm) (*Figure 2F and G*, *Figure 2—figure supplement 5B*, *Video 3*). The thickness of the fluid boundary layer correlated inversely with body speed (*Figure 2H*). These measurements demonstrate shedding of the fluid boundary layer during contractions and significant reduction in the thickness of the fluid boundary layer that had developed during the inter-contraction interval. Thus, recurrent spontaneous contractions result in regular shedding of the fluid boundary layer (as illustrated by the streak of dye left behind in *Figure 2D*), enabling *Hydra* to partially escape from its previous fluid environment and thereby transiently reshape the chemical microenvironment at the epithelial surface where the microbial symbionts are localized. Importantly, the contraction events form a brief perturbation of the inter-contraction interval flow regime at the oral region, while the foot region remains motionless even during peak contractions (*Figure 2B*) and experiences slow flow speeds of maximal values on the order of 0.1 mm/s (*Figure 2—figure supplement 5B*). These flow speeds are comparable to flow speeds experienced at the head between contractions (*Figure 2—figure supplement 5A*). Taken together, these results suggest that *Hydra*'s spontaneous contractions lead to a maximal shedding of viscous boundary layers near the head's surface, and minimal shedding near the foot's surface. This differentiation in the fluid environment could be relevant because of our finding that *Hydra*'s surface is colonized by foot- and head-dominating microbiota (*Figure 1D and E*). Contraction-induced fluid boundary layer shedding may enhance the transport of bacteria-relevant compounds, such as metabolites, antimicrobials, and extracellular vesicles (*Fischbach and Segre, 2016*; *Ñahui Palomino et al., 2021*), between *Hydra*'s head and the fluid environment, compared to lower transport near the foot, hence generating biochemical microhabitats that could promote the observed microbial biogeography. This hypothesis is not readily amenable to experimental interrogation. To date, there are no universal tools for experimentally identifying individual and combinations of the many chemical compounds released or absorbed by microbes (*Thorn and Greenman, 2012*). We therefore probed this hypothesis indirectly. We developed a simple mathematical model of chemical transport in a fluid environment that gets regularly reset by shedding events, as discussed next.

## Physics-based model predicts that contractions increase the exchange rate of chemical compounds to and from the surface

We formulated a simple physics-based mathematical model of the transport of chemical compounds to and from *Hydra*'s surface where the microbes reside. We exploited two key findings from our experimental data. First, between contractions, transport of chemical compounds to and from Hydra's surface is best described by molecular diffusion rather than fluid advection. Second, peak contractions are much shorter and faster than movement during inter-contraction intervals ($T_{PC} \ll T_{IC}$ and $U_{PC} \gg U_{IC}$, see *Figure 2C*), and each peak contraction causes intermittent shedding of the fluid boundary layer and re-extension of *Hydra*'s head in a random direction. Thus, each contraction resets the fluid microenvironment and replenishes the chemical concentration around *Hydra*'s head, while the fluid environment at *Hydra*'s foot remains always at rest, akin to a permanent inter-contraction interval state.

Our dye visualization and flow quantification showed no noticeable background flows between contraction events (*Figure 2D*). The fact that diffusion is dominant between contractions can be formally shown by computing the Péclet number Pe = $LU/D$, which compares the relative importance of advection versus diffusion for the transport of a given compound, such as oxygen. Using the length $L$ of *Hydra*'s body column, the mean flow speed $U$ over 1 hr (averaging over both inter-contraction intervals and contracting periods), and the constant oxygen diffusion D at 15°C, we find that Pe is $0.005 \ll 1$, implying that diffusion is dominant between contraction events, even when the flow speed $U$ is overestimated by averaging over both inter-contraction intervals and spontaneous contraction periods.

To reflect the different fluid microenvironments in the highly motile head and static foot, we approximated the respective head and foot surfaces by two non-interacting spheres of radii $a$ and $b$ separated by a distance $L$ (*Figure 3A*, box). When even a few percent of *Hydra*'s head or foot surface are covered by living microbes or host cells, the diffusion-limited rate of absorption or emission of a chemical compound by these cells is well approximated by a uniformly covered surface (*Berg and Purcell, 1977*). Assuming equivalence of the transport of emitted and absorbed chemical substances (see 'Materials and methods'), we focus here on absorption only. This implies zero concentration of the chemical compound of interest, say oxygen, at *Hydra*'s surface. Starting in a compound-rich environment, a concentration boundary layer (CBL) depleted of that chemical compound forms and grows near the surface. In the absence of contractions, the concentration field reaches a steady state with zero rate of change of the compound concentration. Each spontaneous contraction event sheds the chemically depleted fluid boundary layer near the head and effectively resets the depletion zone growth process (*Figure 3A*). Accounting for unsteady diffusion following each spontaneous contraction, we computed the growth of the depletion boundary layer over time (stages 1–3 in *Figure 3B*), using as an example the diffusion coefficient $D$ of oxygen to derive a dimensional timescale $\tau = a^2/D$ (see 'Materials and methods'). In the absence of fluid boundary layer shedding, such as near *Hydra*'s static foot, steady state is approached as time increases (stage 4 in *Figure 3B*). Near *Hydra*'s head, however, each spontaneous contraction resets the CBL thickness to zero (*Figure 3C*, top), resulting in an instantaneous and increased uptake of molecules, such as oxygen, in the head region (blue) compared to the foot (orange). Consecutive contractions increase the maximal cumulative uptake of compounds near *Hydra*'s head (blue) compared to the foot (orange) (*Figure 3C*, bottom).

To investigate the functional implications of experimentally observed temporal distribution of spontaneous contractions over many hours, we combined our physics-based model of chemical transport (*Figure 3A*) with a stochastic model of contraction events. Specifically, the temporal sequence of *Hydra*'s spontaneous contractions can be estimated mathematically by a Poisson distribution of mean $\lambda$, given that the distribution of inter-contraction intervals are best fitted by an exponential distribution of mean $1/\lambda$ (*Figure 2—figure supplement 3B*). We calculated analytically the expected mean and standard deviation of the cumulative uptake $J_{IC}$ over a single inter-contraction period $T_{IC}$ and of the cumulative uptake J over an extended period T (containing multiple $T_{IC}$) (see 'Materials and methods'). We found that the expected mean value of $J_{IC}$, given by $\langle J_{IC} \rangle = \sqrt{(a^2/\lambda D)}$, decreases with increasing contraction frequency $\lambda$, while the expected mean value of J, given by $\langle J \rangle = \sqrt{(a^2 \lambda/D)}$, increases with increasing $\lambda$. This is intuitive: as the contraction frequency increases, the inter-contraction time $T_{IC}$ decreases, so does the expected uptake $J_{IC}$ over a single inter-contraction period $T_{IC}$. However, the fluid environment gets reset more often, with each resetting event replenishing the chemical concentration, leading to an increase in the expected cumulative uptake $J$ over an extended period $T$ containing multiple contraction events.

Next, we numerically simulated data sets of inter-contraction intervals $T_{IC}$ (*Figure 2—figure supplement 3C*) drawn from exponential distributions of mean $1/\lambda$, where we let $\lambda$ range from 0 to 10 at 0.05 intervals. For each $\lambda$, we conducted 10,000 numerical experiments, each lasting for a fixed total time period $T = 48$ hr. The obtained number of contraction events in each numerical experiment consisted of one realization taken from a Poisson distribution of mean $\lambda$. The total number of contraction events over all experiments was normally distributed, as expected from the law of large numbers (see 'Materials and methods'). We computed numerically the cumulative oxygen uptake J over $T = 48$ hr for each realization (normalized by the cumulative uptake at steady state), and for each $\lambda$, we calculated the mean and standard deviation of the computed $J$. Plotting the mean and standard deviation of $J$ as a function of $\lambda$ (*Figure 3D*), we found that increasing $\lambda$ increases the cumulative uptake

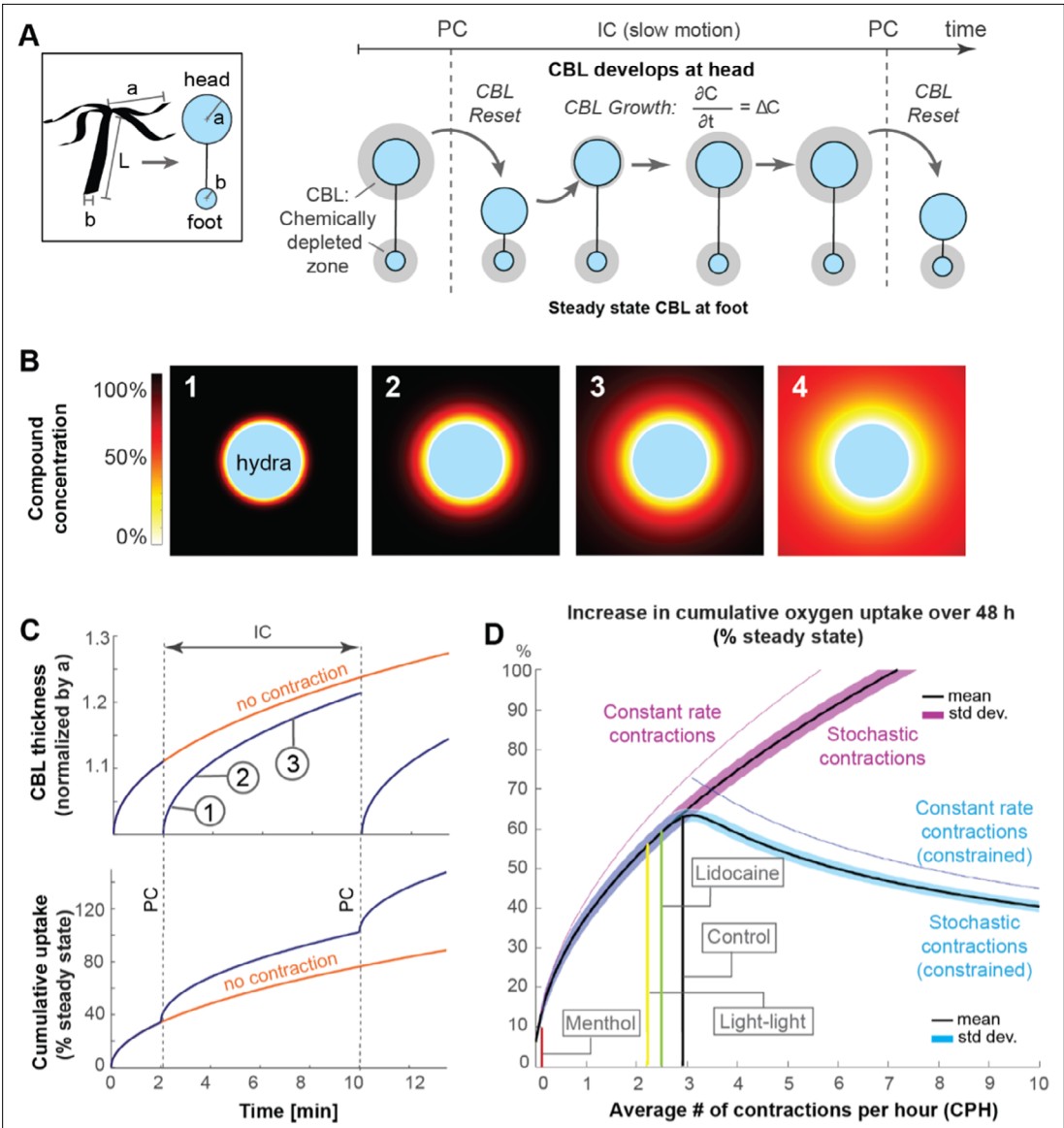

**Figure 3.** Mathematical model suggests that spontaneous contractions enhance mass transport to and from the surface. (**A**) Left, box: simplified animal geometry assumed in model consists of a sphere representing the head and a smaller sphere representing the basal foot. Right: by Fick's law, a chemically depleted concentration boundary layer (CBL) forms around *Hydra*'s head and foot region through continuous uptake of chemical compounds at the surface during the inter-contraction interval (IC). The CBL is shed from the head region, but not the foot region, through spontaneous contractions (PC). (**B**) Computationally modeled growth of chemically depleted CBL around *Hydra*'s head over time until steady state is reached (panel 4), assuming continuous uptake at surface and unlimited supply at far distance. (**C**) Top: growth of chemically depleted CBL as a function of time with and without contractions. Bottom: instantaneous and cumulative uptake rate, respectively, of a given chemical compound (here: oxygen) during the CBL dynamics above; (**D**) Predicted increase in cumulative uptake rate (as percentage of steady state) of a given chemical compound over 48 hr as a function of spontaneous contraction frequencies within the observed range (purple), and of increasing frequencies with unconstraint number of spontaneous contractions (magenta) compared to a constraint number of spontaneous contractions (blue graph).

*J* and that the mean (solid black line) and standard deviation (thick purple segment) are in excellent agreement with analytical predictions (see 'Materials and methods').

Spontaneous contraction frequencies on the order of 10 contractions per hour and more have been reported in the *Hydra* (**Murillo-Rincon et al., 2017**, **Yamamoto and Yuste, 2020**), indicating that, in theory, increases in uptake rates are possible. In our experience, however, such high frequencies occur only transiently when *Hydra* has been stressed, for example, by transferring the animals into very small containers such as concave glass for imaging purposes. At later time points, the contraction

frequency of animals in these conditions converge towards an spontaneous contraction rate around three contractions per hour (*Figure 2—figure supplement 6*), similar to what we observed in our regular setup condition using a large beaker setup (*Figure 2—figure supplement 2A*, control condition). This suggests that under unconstrained conditions, *Hydra*'s baseline contraction rate per hour is near 3, rather than 10.

We used the model to test the effect of limiting the maximal number of contractions over 48 hr to 144 contractions, which is the total number of contractions at an average spontaneous contraction frequency $\lambda = 3$ contractions per hour. Effectively, this constraint means that once the maximum number of contractions is reached, no additional contractions are permitted during the remainder of the 48 hr, mimicking, for example, a finite energy budget for spontaneous contractions. Interestingly, imposing this constraint showed that maximal uptake gains are achieved at contractions frequencies consistent with the imposed constraint (*Figure 3D*, thick blue graph), and that increasing $\lambda$ beyond three contractions per hour decreases the cumulative uptake.

When testing the effect of a constant-rate contraction activity, that is, with a constant $T_{\mathrm{IC}}$ approximately equal to $48/\lambda$ hours, the model predicts a slightly increased uptake compared to stochastic activity, indicating that a precise rhythm would only confer a small benefit over the stochastic mechanism (*Figure 3D*, thin purple and blue graphs). Analogous results hold for the removal for accumulated chemicals produced by microbes at *Hydra*'s surface (see 'Materials and methods').

Our model indicates that *Hydra*'s spontaneous contractions facilitate a greater exchange rate – including both uptake and release – of chemical compounds in the oral region as compared to the static foot region. When spontaneous contraction frequency is reduced, the maximal exchange rate near the head is reduced as well and, in the extreme case of zero contractions, becomes almost identical to the steady state in the foot region.

The simplicity of the model should not distract from the universality of the mechanism it probes: the effect of stochastic contractions on the transport of chemicals to and from a surface exhibiting spontaneous wall contractions. To make analytical progress, we assumed the surface is spherical, but, by continuity arguments, the conclusions we arrived at are qualitatively valid even for non-spherical surfaces. These conclusions can be restated concisely as follows: fast spontaneous contractions of an otherwise slowly moving surface in a stagnant fluid medium cause impromptu shedding of the fluid boundary layer and lead to improved transport of chemicals to and from the surface between contraction events. Higher contraction frequencies are beneficial, but require additional, may be prohibitive, metabolic cost. A stochastic distribution of contraction events over time, following a Poisson process, produces benefits that are nearly as good as those produced by regular contractions, but without the need for a biological machinery to maintain a precise rhythm. Limiting the total number of contractions leads to decreased performance at contraction frequencies beyond what would allow the contraction events to be Poisson distributed.

Taken together, our model suggests that changing the spontaneous contraction frequency will alter the fluid microenvironment and biochemical concentrations experienced by *Hydra*'s microbial community; in particular, reducing the spontaneous contraction frequency would make the microenvironment near *Hydra*'s head, where the greatest fluid boundary layer shedding occurs during spontaneous contractions, more similar to the foot, where minimal fluid boundary layer shedding occurs.

## Reducing the spontaneous contraction frequency changes the colonizing microbiota

To directly probe the impact of spontaneous contractions on the associated microbial community, we decreased *Hydra*'s spontaneous contraction frequency by two established methods: (1) continuous exposure to either light or darkness (*Kanaya et al., 2019*, *Rushforth et al., 1963*) and (2) chemical interference with the ion channel inhibitors menthol and lidocaine (*Klimovich et al., 2020*; *Figure 2—figure supplement 1*). Continuous exposure to light and treatment with ion channel inhibitor lidocaine reduced the spontaneous contraction frequency slightly from an average value of 2.5 contractions per hour in control conditions to 2.3 contractions per hour in continuous light and to 2.0 contractions per hour in lidocaine treatment (*Figure 2—figure supplement 2A*) (note that we were unable to make the measurements in the continuous dark condition). Treatment with ion channel inhibitor menthol almost completely abolished the occurrence of contractions. The treatments reduced spontaneous contraction frequency without significantly changing the frequency of other common fast contractile

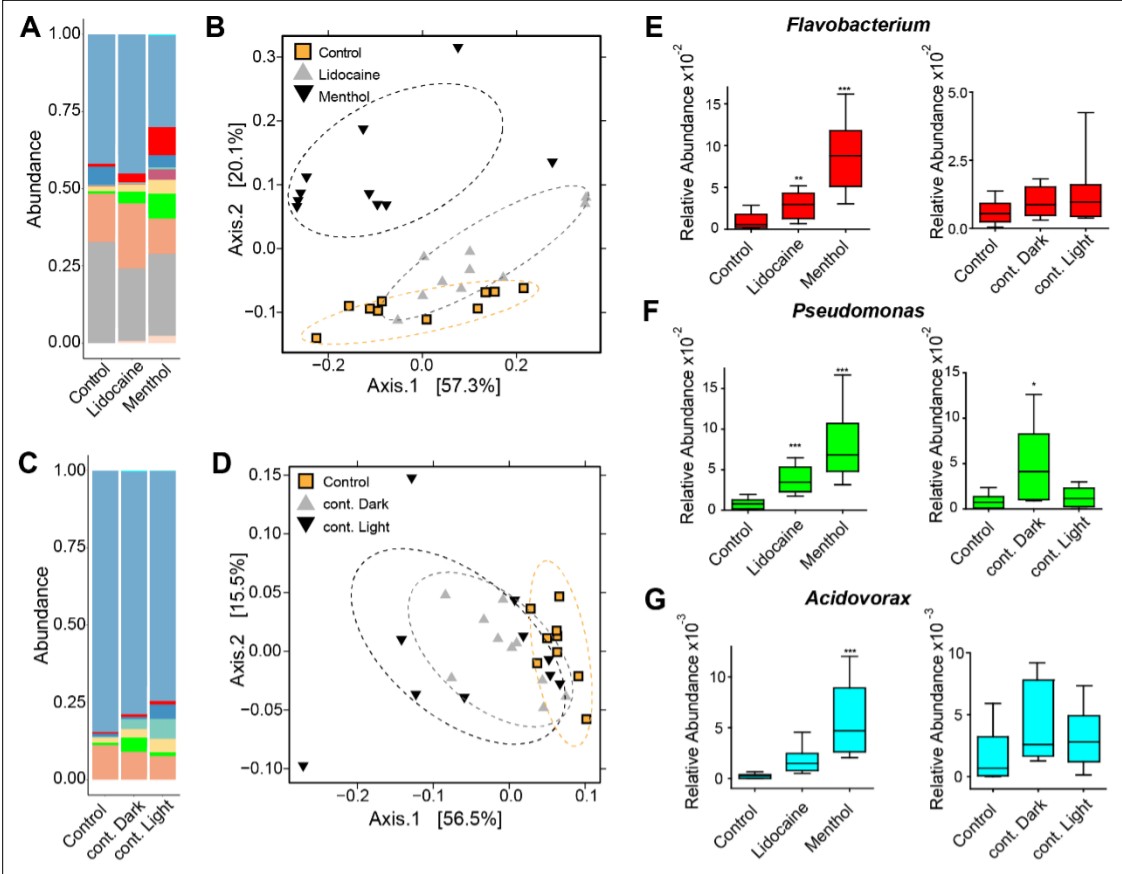

**Figure 4.** Perturbing the frequency of spontaneous contractions over extended time periods shifts the microbial composition in *Hydra*. Reducing the spontaneous contraction frequency with 48 hr treatment of ion channel inhibitors (menthol and lidocaine), continuous light exposure (cont. light), or continuous dark (cont. dark) exposure significantly affects the bacterial community. Control is incubation in freshwater (*Hydra* medium) and 12 hr of light alternating with 12 hr of dark conditions. (**A**) Bar plot of the relative abundance on the genus level showing the effect of the ion channel inhibitors. (**B**) Bar plot of the relative abundance on the genus level showing the effect of the light treatments. (**C, D**) Analysis of the bacterial communities associated with the control and disturbed conditions using principal coordinate analysis of the Bray–Curtis distance matrix. The polyps with disturbed contraction frequency show distinct clustering (ellipses added manually). (**E–G**) Box plots displaying the fold change of the relative abundance compared to the control of *Flavobacterium, Pseudomonas,* and *Acidovorax* in response to the different treatments. (For all plots: n = 8-10) *p≤0.05; **p≤0.01; ***p≤0.001 (ANOVA and Kruskal–Wallis).

The online version of this article includes the following source data and figure supplement(s) for figure 4:

**Figure supplement 1.** The microbiota changes due to reduced contraction frequency.

**Figure supplement 2.** Analysis of the bacterial communities using different distance matrixes.

**Figure supplement 2—source data 1.** Statistical analysis of the PCoAs using Anosim and Adonis.

**Figure supplement 3.** The bacterial load is not affected by ion channel inhibitors or light treatments.

**Figure supplement 4.** Video recording of fluorescently labeled *Curvibacter* reveals a very stable colonization pattern during a full contraction cycle.

**Figure supplement 5.** Reduction in contraction frequency does not alter the glycocalyx of *Hydra*.

behaviors, such as somersaulting (*Figure 2—figure supplement 2B and C*; *Han et al., 2018*). Consistent with earlier studies (*Kanaya et al., 2019*, *Klimovich et al., 2020*), lidocaine and constant light treatment increased the likelihood of longer TIC; the contraction events remained, however, Poisson-distributed and TIC remained exponentially distributed (*Figure 2—figure supplement 3B*), as assumed in our mathematical model. Computing the average contraction frequency based on the best exponential fit to biological data, we found that it to be equal to 2.5, and 2.2 contractions per hour for the lidocaine, and continuous light treatments, respectively, as opposed to 2.9 contractions per hour for the control, again confirming the reduced contraction rate in the treatment groups.

We next investigated the effects of the altered spontaneous contraction frequency on *Hydra*'s microbiota. A 48 hr exposure to either pharmacological agents (menthol and lidocaine) or exposure to different light regimes led to a similar and significant shift in the microbial community (*Figure 4*, *Figure 4—figure supplement 1*). The relative abundance bar plot and principal coordinate analysis (PCoA) of the microbial composition showed a clustering of the different treatments in response to a disturbed contraction frequency. The extent of the shift was positively correlated with the magnitude of the decrease in contraction frequency and, consequently, was the greatest in menthol-treated animals. This shift consisted of a change in the relative abundance rather than a disappearance of bacterial taxa or bacterial load (*Figure 4—figure supplement 2*). The fact that both light and pharmacological manipulations resulted in similar effects provides strong evidence that the change in relative bacterial abundance is related to the altered spontaneous contraction frequencies. We further verified by using a minimal inhibitory concentration (MIC) assay that there were no inhibitory effects of the pharmacological substances on the bacteria (*Figure 4—figure supplement 3*).

We evaluated the differences between control and treatment groups for up- or downregulated bacteria using linear discriminant analysis effect size (LEfSe) (*Segata et al., 2011*). When spontaneous contractions were reduced, the foot-specific microbial colonizers *Flavobacterium, Pseudomonas,* and *Acidovorax* (*Figure 1E and F*) became more abundant (*Figure 4E–G*). Taken together, these data suggest that a reduced spontaneous contraction frequency alters the microbial composition and potentially expands the biogeography of foot-associated bacteria.

Our fluid and transport analysis suggests that these changes in the microbiota result from an altered biochemical microenvironment near *Hydra*'s head, which is shaped by the spontaneous contraction frequency. In particular, our mathematical model, which faithfully mimics the distribution of inter-contraction intervals under control and disturbed conditions (*Figure 2—figure supplement 3B and C*), predicts that the reduction in spontaneous contraction frequency seen in the experimental treatments (lidocaine, methanol, continuous light) decreases the cumulative uptake or release of any bacteria-relevant chemical compound near *Hydra*'s head (*Figure 3C*), such that the head's biochemical microenvironment becomes more similar to the static foot region (*Figure 3C*).

However, another explanation of the observed changes in the microbiota could be that the contractions are important for the direct displacement of the microbes to other parts of the body. To investigate whether spontaneous contraction have a direct physical impact on the microbiome, we first took video recordings of a normal *Hydra* polyp colonized with fluorescently labeled bacteria during a full contraction cycle. We could neither observe an overall change in the colonization pattern nor the physical detachment of labeled bacteria (*Figure 4—figure supplement 4*, *Videos 4 and 5*). Thus, at short time scale (few seconds), an individual contraction cycle does not affect the biogeography of the microbiome. We also probed whether a lack of recurrent

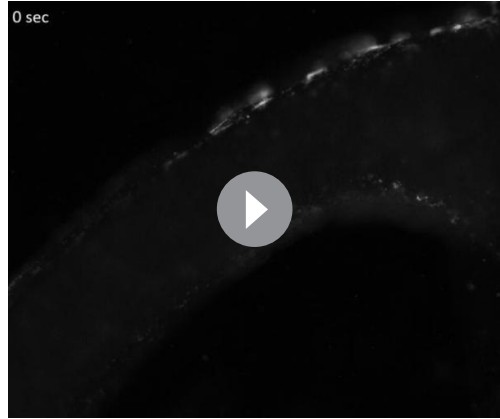

**Video 5.** *Hydra*'s body column during a full-body contraction colonized with fluorescent *Curvibacter* sp. AEP1.3 (white signal on the surface). https://elifesciences.org/articles/83637/figures#video5

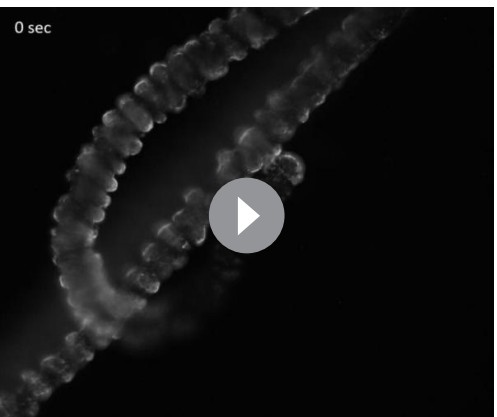

**Video 4.** *Hydra*'s tentacles during a full-body contraction colonized with fluorescent *Curvibacter* sp. AEP1.3 (white signal on the surface). https://elifesciences.org/articles/83637/figures#video4

spontaneous contractions may lead to a remodeled glycocalyx, for example, by hindering its redistribution during spontaneous contractions, and thereby alter the growth conditions of bacteria. Here, we used an antibody specific for a component of *Hydra*'s glycocalyx (*Böttger et al., 2012*) and visualized the glycocalyx in animals immobilized for a prolonged (48 hr) time period (*Figure 4—figure supplement 5*). Since we could not detect major changes in the glycocalyx layer, we concluded that the observed changes in the microbiota are unlikely the cause of glycocalyx remodeling. These results strongly suggest that the fluid boundary layer shedding and resultant chemical exchange by *Hydra*'s spontaneous contractions stabilize microbial colonization and contribute to shaping the microbial biogeography along the body column (*Figure 1D and E*).

## Discussion

The epithelia and microbial symbionts of *Hydra*'s body walls interface with slow-moving or highly viscous fluids. They rely on diffusion for the transport of chemical compounds to and from the surface. *Hydra* cannot engage in more efficient transport mechanisms via fluid advection (*Nawroth et al., 2017*, *Gilpin et al., 2017*, *Pepper et al., 2010*) because it lacks the ciliary or muscular appendages used by other organisms in the low and intermediate Reynolds number regimes to generate unsteady flows and vortices. Soft corals, for example, operate at intermediate Reynolds numbers and use rhythmic pulsations of their tentacles to generate bulk flow that removes oxygen waste (*Samson et al., 2019*). Although individual corals reach diameters similar to that of fully extended *Hydra* (ca. 1 cm), they usually occur in dense clusters and contract rhythmically at 0.5 Hz, leading to collective fluid advection with Péclet numbers on the order of 10–1000, compared to Pe = 0.005 computed for a single *Hydra* with only a handful of contractions per hour, indicating that the two organisms represent different scenarios. Our study suggests that *Hydra* does not leverage fluid advection but instead facilitates the diffusion process through its spontaneous contractions, which cause transiently and locally higher flow rates and greatly improve the exchange of fluid near the epithelial surface by shedding the so-called fluid boundary layer. This is analogous to a recently discovered 'sneezing' mechanism by which marine sponges remove mucus from their surface by recurrent spontaneous contractions (*Kornder et al., 2022*). Combining high-throughput microbiota profiling, in vivo flow analysis, and mathematical modeling, we found that such exchange facilitates the transport of compounds, including nutrients, waste, antimicrobials, and gases, to and from surface-residing symbiotic microbiota and may take part in maintaining a specific microbiota biogeography along the *Hydra*'s body column (*Augustin et al., 2017*). Interestingly, this distinct spatial colonization pattern along the body column could not only be essential for pathogen defense (*Bosch, 2013*, *Fraune et al., 2015*) but could also be important for modulating the frequency of *Hydra*'s spontaneous contractions (*Murillo-Rincon et al., 2017*), suggesting a feedback loop that could be the focus of future studies.

Our results may also have broader applicability. *Hydra*'s mucous layer covering the epithelium – an inner layer with stratified organization devoid of bacteria, beneath an outer loose layer colonized by symbionts – is similar to the mammalian colon. Like *Hydra*, the peristaltic gut displays recurrent spontaneous contractions (*Gill et al., 1986*, *Koch et al., 1988*, *Swaminathan et al., 2016*) and a viscosity-dominated, low Reynolds number flow regime (*Janssen et al., 2007*). Disturbance of this contractile activity (e.g., intestinal dysmotility) is correlated with dysbiosis (*Hadizadeh et al., 2017*, *Scott and Cahall, 1982*, *Vandeputte et al., 2016*), which can lead to bacterial overgrowth in the small intestine and irritable bowel syndrome (*Kostic et al., 2014*, *Toskes, 1993*). Microfluidic models mimicking the viscous environment and laminar flow of the gut suggest that intestinal contractions mix the luminal content, which appears to modulate microbial density and composition (*Cremer et al., 2016*). These observations are consistent with our findings in *Hydra* and suggest that spontaneous contractions might both depend on microbial colonizers and also regulate host–bacteria associations.

Finally, our non-dimensional model enables the generalized prediction that body wall contractions, whether in *Hydra*, the gut, or other systems, cause relative improvement of transport for any compound that is produced or consumed near an epithelial surface in low Reynolds number conditions, thereby simultaneously preventing the build-up of metabolic waste and restoring depleted nutrients. Lowering the frequency of spontaneous contractions by 10–15%, as seen in our experiments, is predicted to reduce the uptake (or removal) rates of individual compounds by a similar order of magnitude. It is intriguing that such a relatively small change in spontaneous contraction rate altered the microbiome significantly and consistently across treatments. One explanation is that since

fluid boundary layer shedding by contractions affects the transport of any chemical compound near the surface, decreasing the contraction rate might alter the uptake or removal of many compounds at once. As shown in a recent study on the response of microbial populations to altered soil properties, such combinatorial effects can greatly exceed, and complicate, the impact of any single factor (*Rillig et al., 2019*). Intuitively, and confirmed by our model, increasing the frequency of contractions results in greater transport to and from the surface. Since spontaneous contractions are energetically expensive processes, *Hydra* under normal conditions may get along well with a moderate rate of about three contractions per hour. Our computational model indicates that this rate still provides significant enhancement over purely diffusive transport, and our experimental data show that lower rates tend to destabilize the microbiome. Taken together, this suggests that *Hydra*'s rhythm of spontaneous contractions might be operating at a sweet spot of efficiency. Furthermore, we demonstrate that a stochastic distribution of a limited number of contractions over time results in a more effective transport than other strategies, such as alternating periods with a higher frequency of contractions with periods that are static. While contractions at regular time intervals would be slightly more effective still, this would require the presence and maintenance of pacemaker circuits with precisely controlled firing frequency. Though future work is needed, our results suggest that *Hydra* – and possibly other contractile epithelial systems – achieve near optimal efficiency without the need for a costly high-precision pacemaker system.

Inspired by Dobzhansky's dictum that "nothing in biology makes sense but in the light of evolution" (*Dobzhansky, 1964*), we speculate that spontaneous contractions were critical early in the evolution of a gut system to maintain a stable microbiota. In this context, an observation in a new species of hydrothermal vent Yeti crabs, *Kiwa puravida* n.sp. is of interest (*Thurber et al., 2011*). The crabs farm symbiotic bacteria on the surface of their chelipeds (claws) and wave these chelipeds continuously in a constant rhythm. In search of an explanation, the authors stated: "We hypothesize that K. puravida n. sp. waves its chelipeds to shear off boundary layers formed by their epibionts productivity, increasing both the epibionts and, in turn, their own access to food" (*Thurber et al., 2011*). Pointing in the same direction, a study on multiciliated surface cells in amphibian and some fish embryos ends with the hypothesis that the fluid flow generated by these cells may contribute to managing embryo-associated microbial consortia (*Kerney, 2021*). In light of these interesting hypotheses, well-controlled perturbation experiments in *Hydra* may offer a unique model to study both the mechanisms and the in vivo role of host generated water flow in maintaining a stable microbiome.

## Materials and methods
### Animal manipulation and data analysis
#### Animal culture
Experiments were carried out using *Hydra vulgaris* strain AEP. Animals were maintained under constant environmental conditions, including culture medium (*Hydra* medium; 0.28 mM CaCl$_2$, 0.33 mM MgSO$_4$, 0.5 mM NaHCO$_3$, and 0.08 mM KCO$_3$), temperature (18°C), and food according to standard procedures (*Klimovich et al., 2019*). Experimental animals were chosen randomly from clonally growing asexual *Hydra* cultures. The animals were typically fed three times a week with first-instar larvae of *Artemia salina*; however, they were not fed for 24 hr prior to pharmacological interference or light assays, or for 48 hr prior to RNA isolation.

#### Pharmacological interference and light assays
To alter the contraction frequency of *H. vulgaris* AEP polyps, the animals were treated for 48 hr with either 200 µM menthol (Sigma, Cat# 15785) or 100 µM lidocaine (Sigma, Cat# L5647), or exposed to 48 hr of continuous light or darkness. Control polyps were incubated in *Hydra* medium and were exposed to a 12 hr light/12 hr dark cycle over the same time period. Water temperature remained stable at 18°C for all conditions. From hour 24 to hour 32 of the 48 hr experiment, that is, for 8 hr total, 10 polyps each were simultaneously video-recorded in a 50 mL glass beaker, using a frame rate of 20 frames per minute. For comparison with prior studies, single polyps were recorded for 8 hr in a concave slide with a medium volume of 200–500 µL as previously described (*Murillo-Rincon et al., 2017*). Using the video recordings, we quantified the number of full-body contractions and

somersaulting events in ImageJ/Fiji (*Schindelin et al., 2012*) and computed the average contraction frequency per hour.

## DNA extraction and 16S rRNA profiling

To investigate the effects of a reduced contraction frequency on the microbiota, we exposed normal *H. vulgaris* AEP polyps to different pharmacological agents and light conditions over a 48 hr time period. Polyps were treated with 200 µM menthol (Sigma, Cat# 15785), 100 µM lidocaine (Sigma, Cat# L5647), continuous light or continuous darkness for 48 hr at 18°C. Control polyps were incubated in *Hydra* medium and were exposed to a 12 hr light/12 hr dark cycle. Afterward the genomic DNA was extracted from individual polyps with the DNeasy Blood and Tissue Kit (QIAGEN) as described in the manufacturer's protocol. Elution was performed in 50 µL. Extracted DNA was stored at –20°C until sequencing. Prior to sequencing, the variable regions 1 and 2 (V1V2) of the bacterial 16S rRNA genes were amplified according to the previously established protocol using the primers 27F and 338R (*Rausch et al., 2016*). For bacterial 16S rRNA profiling, paired-end sequencing of 2 × 300 bp was performed on the Illumina MiSeq platform. The 16S rRNA sequencing raw data are deposited at the SRA and are available under the project ID PRJNA842888. The sequence analysis was conducted using the DADA2 pipeline in R 3.6.0 (*Callahan et al., 2016*). The downstream analysis of the 16S rRNA data (alpha diversity, relative abundance, and beta-diversity) was done in R including the packages phyolseq, vegan, DESeq2, and ggplot2 (*McMurdie and Holmes, 2013*; *Oksanen et al., 2013*; *Love et al., 2014*; *Ginestet, 2011*). ASVs with <10 reads were removed from the data set. The tables of bacterial abundance were further processed using LEfSe analysis to identify bacterial taxa that account for major differences between microbial communities (*Segata et al., 2011*). An effect size threshold of 3.0 (on a log10 scale) was used for all comparisons discussed in this study. The results of LEfSe analysis were visualized by plotting the phylogenetic distribution of the differentially abundant bacterial taxa on the Ribosomal Database Project (RDP) bacterial taxonomy.

## Statistics

Statistical analyses were performed using two-tailed Student's *t*-test or Mann–Whitney *U*-test where applicable. If multiple testing was performed, p-values were adjusted using Bonferroni correction.

## Quantitative real-time PCR analysis (qRT-PCR)

In order to investigate the bacterial community change and validate if there is an increase in the bacterial load, we performed quantitative real-time PCR analysis. The samples from the 16S rRNA profiling experiment were used. Amplification was performed as previously described (*Klimovich et al., 2018b*) using GoTaq qPCR Master Mix (Promega, Madison, USA) and specific oligonucleotide primers (EUB 27F, EUB 338R). Also, 4–5 biological replicates of each treatment (light, darkness, menthol, and lidocaine) and control with two technical replications were analyzed. The data were collected using ABI 7300 Real-Time PCR System (Applied Biosystems, Foster City, USA) and analyzed by the conventional ΔΔCt method.

## Minimal inhibitory concentration assay (MIC)

To test whether the ion channel inhibitors (menthol and lidocaine) have an antimicrobial activity, their effect was tested in the MIC assays as previously described (*Augustin et al., 2017*). The following bacterial strain isolates from the natural *H. vulgaris* strain AEP microbiota were used: *Curvibacter* sp., *Acidovorax* sp., *Pelomonas* sp., *Undibacterium* sp., and *Duganella* sp. (*Franzenburg et al., 2013*; *Klimovich et al., 2020*). Microdilution susceptibility assays were performed in 96-microtiter-well plates. We tested a range of concentrations of the ion channel inhibitors in *Hydra* medium (with an additional 10% concentration of R2A agar) to match the nutrient-poor conditions of the behavioral assay. The concentration range was chosen such that the dose used in the behavioral assay was the middle of the dilution series. The following concentration were tested (in µM): menthol: 400, 300, 200, 100, and 10; and lidocaine: 10,000, 5000, 2500, 1000, and 100. The inoculum of approximately 100 CFU per well was used. The plates were incubated with the inhibitors for 5–7 d at 18°C. The MIC was determined as the lowest concentration showing the absence of a bacterial cell pellet. The run was designed in a way that every concentration had four replicates.

## Immunochemical staining of the glycocalyx

To test whether alterations in the contraction frequency has any effect on the glycocalyx of *Hydra*, we incubated polyps for 48 hr in menthol solution or S-medium (control) and visualized the glycocalyx by immunochemical staining and confocal microscopy of whole-mount polyps. To facilitate the detection of the *Hydra*'s epithelial surface, we used transgenic polyps expressing eGFP in the ectoderm (ecto-GFP line A8; *Wittlieb et al., 2006*). The glycocalyx was stained using a polyclonal antibody raised in chicken against the PPOD4 protein, generously provided by Prof. Angelika Böttger. The PPOD4 protein has been previously shown to specifically localize to the glycocalyx layer adjacent to the membrane of ectodermal epithelial cells in *Hydra* (*Böttger et al., 2012*). Polyclonal rabbit-anti-GFP antibody (AB3080, Merck) was used to amplify the GFP signal. Immunohistochemical detection was carried out as described previously (*Klimovich et al., 2018b*). Briefly, polyps were relaxed in ice-cold urethane, fixed in 4% paraformaldehyde, incubated in blocking solution for 1 hr, and incubated further with the primary antibodies diluted to 1.0 µg/mL in blocking solution at 4°C. Following the protocol of Böttger and co-authors, tissue permeabilization steps were omitted to avoid detection of immature glycocalyx components within epithelial cells. Alexa Fluor 488-conjugated goat anti-rabbit antibodies (A11034, Thermo Fisher) and Alexa Fluor 546-conjugated goat anti-chicken antibodies (A11040, Thermo Fisher) were diluted to 2.0 µg/mL and incubations were carried out for 2 hr at room temperature. The samples were mounted into Mowiol supplemented with 1.0% DABCO antifade (D27802, Sigma). Confocal mid-body optical sections were captured using a Zeiss LSM900 laser scanning confocal microscope. To measure the glycocalyx thickness, the fluorescence profile across the ectoderm has been recorded and quantified for 10 transects, each 10 µm long, using Zen Blue v. 3.4.91 software (Zeiss).

## GFP labeling of *Curvibacter* sp. AEP1.3

To visualize the colonization and dynamics during contraction events of *Curvibacter* sp. AEP1.3, we chromosomally integrated sfGFP behind the glmS-operon via the miniTn7-system as previously established by *Wiles et al., 2018*. The protocol was modified in order to manipulate the freshwater bacterium *Curvibacter* sp. AEP1.3 as follows: instead of the *Escherichia coli* SM10, we used the strain *E. coli* MFDpir as delivery system because bi- and triparental mating was already observed (*Wein et al., 2018*; *Ferrières et al., 2010*). In addition, the growth medium was changed to the routinely used medium of *Curvibacter* sp.R2A and antibiotic concentration of the selection media was adjusted to 2 µg/mL.

## Kinematics and fluid flow analysis

### Fluid boundary layer visualization

The fluid boundary layer around *Hydra* was visualized as described previously (*Nawroth et al., 2010*; *Nawroth and Dabiri, 2014*). Briefly, animals were placed into custom-made cube-shaped acrylic containers filled with Hydra medium and allowed to acclimatize for 30 min. The fluid boundary layer was visualized using fluorescein dye (Sigma-Aldrich) added near *Hydra*'s head during inter-contraction periods using a transfer pipette. An LED light source was used to illuminate the dye from the side. Videos were recorded using a Sony HDR-SR12 camcorder (1440 × 1080 pixels, 30 frames per second; Sony Electronics, San Diego, CA) mounted to a tripod in front of the custom container. Videos were processed to enhance the contrast of the dye with respect to the background using Adobe Premiere Pro (San Jose, CA).

### Fluid flow analysis

Fluid flow around *H. vulgaris* AEP polyps was quantified using particle image velocimetry under a stereo microscope as described previously (*Nawroth et al., 2012*). Briefly, *H. vulgaris* AEP polyps were placed in a custom-made acrylic containers with glass walls filled with *Hydra*-medium that contained 1 µm green-fluorescent microspheres (Invitrogen). A violet laser pointer mounted on a custom micromanipulator stage was aligned with a plano-concave cylindrical lens with a focal length of −4 mm (Thorlabs, Newton, MA) to create an excitation light sheet that captured the particles in ca. 1 mm thick plane across or along the animal's body column. The green-emitting particles were recorded from top or side using a stereo microscope equipped with a long pass filter (to remove the

violet excitation wavelength) and a camera (same as above) mounted to the eyepiece. Videos were recorded at 60 frames per second. We used MATLAB (MathWorks, Natick, MA) with an open-source code package (PIVlab *Thielicke and Stamhuis, 2014*) to measure the particle displacement field across the illuminated plane at each time point and derive the instantaneous planar flow velocity field. From this vector field, the flow velocity magnitude (speed) profiles along lines extending perpendicular form *hydra*'s surface (*Figure 2E and F*) were extracted to determine the fluid boundary layer thickness (*Figure 2—figure supplement 5*), defined here as the distance normal from the surface to the point in the fluid where flow velocity has reached 90% of free stream velocity (here assumed zero), as commonly done in biological systems (*Vogel, 2020*).

## Body kinematics

From the videos described above, body kinematics were extracted by manually labeling the head and the foot of *Hydra* in the video and then tracking the displacement of the labels over time using the open-source software Fiji (*Schindelin et al., 2012*) with plugin TrackMate (*Tinevez et al., 2017*). Using these displacement data, we computed *Hydra* body length and movement speed using MATLAB.

## Physics-based modeling of chemical transport to *Hydra*'s surface

### Concentration field surrounding an absorbing surface

We approximated *Hydra*'s head by a sphere of radius $a$ for simplicity and analyzed the case in which only molecular diffusion in the suspending fluid governs transport (uptake) of a chemical species of diffusivity $D$ to *Hydra*'s spherical head. Assuming axisymmetry, the concentration of the chemical compound $C(r, t)$ depends on radial distance $r$ from the center of *Hydra*'s head and $t$ is time. The unsteady diffusion equation in spherical coordinates is given by *Carslaw and Jaeger, 1959*; *Berg and Purcell, 1977*.

$$\frac{\partial C}{\partial t} = D \frac{1}{r^2} \left[ \frac{\partial}{\partial r} \left( r^2 \frac{\partial C}{\partial r} \right) \right], \tag{1}$$

subject to absorbing boundary conditions $C(t, r = a) = 0$ at *Hydra*'s surface and constant concentration far from the *Hydra* $C(t, r \rightarrow \infty) = C_\infty$. The unsteady solution is given by

$$C(t, r) = C_\infty \left[ 1 - \frac{a}{r} + \frac{a}{r} \mathrm{erf} \left( \frac{r-a}{\sqrt{4Dt}} \right) \right]. \tag{2}$$

where $\mathrm{erf}(\cdot)$ is the error function with $\mathrm{erf}(0) = 0$ and $\mathrm{erf}(\infty) = 1$. Note that this solution satisfies the boundary and initial conditions converges as $t \rightarrow \infty$ to the steady-state solution $C = C_\infty(1 - a/r)$.

The radial concentration gradient is given by

$$\frac{\partial C}{\partial r} = C_\infty \left[ \frac{a}{r^2} - \frac{a}{r^2} \mathrm{erf} \left( \frac{r-a}{\sqrt{4Dt}} \right) + \frac{a}{r} \frac{2}{\sqrt{\pi}} \frac{1}{\sqrt{4Dt}} e^{-(r-a)^2/4Dt} \right]. \tag{3}$$

The unsteady uptake of a given chemical compound is defined by $I = -\oint \mathbf{n} \cdot D\nabla C \mathrm{d}S$, where $\mathbf{n}$ is the unit normal, and $\mathrm{d}S = 2\pi R^2 \sin\theta \mathrm{d}\theta$ is the element of surface area of the sphere,

$$I = \int_0^\pi 2\pi D \left. \frac{\partial C}{\partial r} \right|_{r=a} a^2 \sin\theta \mathrm{d}\theta = 4\pi D C_\infty a \left[ 1 + \frac{1}{\sqrt{\pi}} \sqrt{\frac{\tau}{t}} \right], \tag{4}$$

where $\tau = a^2/D$ is the characteristic diffusion time scale.

At steady state, the concentration is given by $C_{ss}(r) = C_\infty(1 - a/r)$; the instantaneous uptake of a chemical compound is $I_{ss} = 4\pi D C_\infty a$, and the cumulative uptake $J_{ss}$ over a time interval $T$ is $J_{ss} = I_{ss}T$.

### Concentration field surrounding an emitting surface

The transport of a chemical compound of diffusivity $D$ produced by emitters covering *Hydra*'s head is equivalent but opposite to that of a compound absorbed at the surface. To verify this statement, let $\tilde{C}(t, r)$ be the concentration field around an emitting surface. The concentration field $\tilde{C}$ is governed by the unsteady diffusion equation in (*Equation 1*), albeit for a different set of boundary conditions. At the surface, the concentration is given by the boundary condition $\tilde{C}(t, r = a) = C_\infty$, and far from

the surface the concentration is nil, $\tilde{C}(t, r \to \infty) = 0$. Introducing the transformation $\tilde{C} = C_\infty - C$, and by linearity of (*Equation 1*), we get that the unsteady solution is $\tilde{C}(t, r) = C_\infty - C(t, r)$, where $C(t, r)$ is given in (*Equation 2*),

$$\tilde{C} = C_\infty \left[ \frac{a}{r} - \frac{a}{r} \text{erf} \left( \frac{r - a}{\sqrt{4Dt}} \right) \right]. \tag{5}$$

The radial concentration gradient $\partial \tilde{C}/\partial r = -\partial C/\partial r$ is the opposite of the absorbing gradient in (*Equation 3*), and the unsteady production of a given chemical compound $\tilde{I} = -I$ is opposite to the unsteady uptake $I$ in (*Equation 4*). It thus suffices to study either chemical absorption or production, with the understanding that the two systems are equivalent. In the following, we focus on chemical absorption.

## Concentration uptake modified by contraction events

Considering a contraction event at $t = 0$ and assuming no additional contractions during an interval $T_i$, corresponding to an inter-contraction time period, the cumulative uptake during this time period becomes

$$J_i = \int_0^{T_i} I dt = 4\pi D C_\infty a \int_0^{T_i} \left( 1 + \frac{1}{\sqrt{\pi}} \sqrt{\frac{\tau}{t}} \right) dt. \tag{6}$$

Upon integration, we get

$$J_i = I_{\text{ss}} T_i \left[ 1 + \frac{2}{\sqrt{\pi}} \sqrt{\frac{\tau}{T_i}} \right]. \tag{7}$$

The change in the cumulative intake over one inter-contraction period $T_i$ relative to the cumulative intake at steady state over the same time period $T_i$ is given by

$$\frac{\Delta J_i}{J_{i\,\text{ss}}} = \frac{J_i - J_{i\,\text{ss}}}{J_{i\,\text{ss}}} = \frac{2\sqrt{\tau T_i}}{\sqrt{\pi}}. \tag{8}$$

If we had $N$ contraction events during a time interval $T$, including the first event at $t = 0$, with inter-contraction time periods $T_i$, $i = 1, \ldots, N$, that are not necessarily equal, such that $T = \sum_{i=1}^N T_i$, then the total uptake during that period would be (noting that $J_{\text{ss}} = I_{\text{ss}} T$)

$$J = 4\pi D C_\infty a \sum_{i=1}^N \left[ \int_0^{T_i} \left( 1 + \frac{1}{\sqrt{\pi}} \sqrt{\frac{\tau}{t}} \right) dt \right] = J_{\text{ss}} + J_{\text{ss}} \frac{1}{T} \sum_{i=1}^N \frac{2\sqrt{\tau T_i}}{\sqrt{\pi}}. \tag{9}$$

The change in the total cumulative intake with multiple contraction events over the total time period $T$ relative to the cumulative intake at steady state over the same time period is given by

$$\frac{\Delta J}{J_{\text{ss}}} = \frac{J - J_{\text{ss}}}{J_{\text{ss}}} = \frac{1}{T} \sum_{i=1}^N \frac{\Delta J_i}{J_{i\,\text{ss}}} = \frac{1}{T} \sum_{i=1}^N \frac{2\sqrt{\tau T_i}}{\sqrt{\pi}}. \tag{10}$$

# Coupling chemical transport with stochastic contraction events

We have established in this work that the number of contraction events in a total time period $T$ is a Poisson process with expectation $\lambda T$ and variance $\sqrt{\lambda T}$, and that the inter-contraction time $T_i$ is exponentially distributed with expectation $1/\lambda$ and variance $1/\lambda$. Here, $\lambda$ is the number of contractions per time unit. Here, we derive analytical expressions of the expectations and variances of the incremental and cumulative uptake of a chemical compound between and across contraction events, respectively.

## Expectation and variance of incremental and cumulative uptake

By definition, the change in cumulative uptake $\Delta J_i / J_{i\,\text{ss}}$ over an interval $T_i$, is a function of the inter-contraction time $T_i$, which is an exponentially distributed random variable. Therefore $\Delta J_i / J_{i\,\text{ss}}$ itself is a random variable. We wish to compute its probability distribution function. Using standard tools from stochastic calculus (*Ross, 2014*), we arrive at the probability distribution function of the random variable $\Delta J_i / J_{i\,\text{ss}}$,

$$P\left(\frac{\Delta J_i}{J_{i\,\mathrm{ss}}} = x\right) = \frac{2\lambda x}{\alpha^2} e^{-\lambda x^2/\alpha^2}, \tag{11}$$

where $\alpha = \frac{2\sqrt{\tau}}{\sqrt{\pi}}$ is a constant introduced for notational convenience.

The expectation of the change in incremental uptake over an interval $T_i$ is given by $\mathbb{E}\left[\frac{\Delta J_i}{J_{i\,\mathrm{ss}}}\right] = \int_0^\infty xP(x)dx$. Upon integration by parts, we arrive at

$$\mathbb{E}\left[\frac{\Delta J_i}{J_{i\,\mathrm{ss}}}\right] = \frac{\sqrt{\tau}}{\sqrt{\lambda}}. \tag{12}$$

The variance is defined as

$$\mathbb{V}\left[\frac{\Delta J_i}{J_{\mathrm{ss}}}\right] = \mathbb{E}\left[\left(\frac{\Delta J_i}{J_{\mathrm{ss}}} - \mathbb{E}\left[\frac{\Delta J_i}{J_{\mathrm{ss}}}\right]\right)^2\right] \qquad \Longrightarrow \qquad \mathbb{V}\left[\frac{\Delta J_i}{J_{\mathrm{ss}}}\right] = \frac{(4-\pi)}{\pi}\frac{\tau}{\lambda}. \tag{13}$$

The change $\Delta J/J_{\mathrm{ss}}$ in cumulative uptake over the time interval $T$ spanning several inter-contraction time periods $T_i$ is also a stochastic random variable defined as the random sum of the random variable $\Delta J_i/J_{\mathrm{ss}}$ over a random number $N$ of contraction events within $T$, given in (*Equation 10*). While a closed-form expression of the probability distribution function of $\Delta J/J_{\mathrm{ss}}$ is not readily available, analytical progress can be made by computing the expectation of $\Delta J/J_{\mathrm{ss}}$

$$\mathbb{E}\left[\frac{\Delta J}{J_{\mathrm{ss}}}\right] = \frac{1}{T}\mathbb{E}\left[\frac{\Delta J_i}{J_{i\,\mathrm{ss}}}\right]\mathbb{E}[N]. \tag{14}$$

Since contraction events follow a Poisson process, the expectation of $N$ contraction events within a time period $T$ is given by $\mathbb{E}[N] = \lambda T$. Therefore, we can express the expectation of the change in cumulative chemical uptake over an interval of time $T$ by

**Table 1.** Random variables.

The number of contraction events $N$ in a time period $T$ follows a Poisson distribution, and the inter-contraction times $T_i$ follow an exponential distribution. Combining this stochastic description of contraction events and inter-contraction times with physics-based models of chemical uptake over the interval $T$, we arrive at a stochastic model of incremental uptake over each inter-contraction period $T_i$ and cumulative uptake over the interval $T = \sum_i^N T_i$. The p.d.f., mean (expectation) and standard deviation (stdev =square root of the variance) of the input $(N, T_i)$ and output $\left(\frac{\Delta J_i}{J_{i\,\mathrm{ss}}}, \frac{\Delta J}{J_{\mathrm{ss}}}\right)$ random variables are listed here.

| | Prob. dist. function | Mean | Stdev |
|---|---|---|---|
| Contraction events | Poisson: $P(N) = \dfrac{(\lambda T)^N e^{\lambda T}}{N!}$ | $\lambda T$ | $\sqrt{\lambda T}$ |
| Inter-contraction time | Exponential: $P(T_i) = \lambda e^{-\lambda T_i}$ | $1/\lambda$ | $1/\lambda$ |
| Incremental uptake | $P\left(\dfrac{\Delta J_i}{J_{i\,\mathrm{ss}}} = x\right) = \dfrac{2\lambda x}{\alpha^2} e^{-\lambda x^2/\alpha^2}$ | $\sqrt{\tau/\lambda}$ | $\sqrt{\dfrac{(4-\pi)}{\pi}\dfrac{\tau}{\lambda}}$ |
| Cumulative uptake | $P\left(\dfrac{\Delta J}{J_{\mathrm{ss}}}\right) = P\left(\dfrac{1}{T}\sum_{i=1}^{N}\dfrac{\Delta J_i}{J_{i\,\mathrm{ss}}}\right)$ | $\sqrt{\lambda\tau}$ | $\sqrt{\dfrac{(4-\pi)\tau}{\pi T}}$ |

$$\mathbb{E}\left[\frac{\Delta J}{J_{ss}}\right] = \sqrt{\tau \lambda} \qquad (15)$$

The variance of $\Delta J/J_{ss}$ conditional on $N$ contractions during a fixed interval $T$ is expressed by

$$\mathbb{V}\left[\frac{\Delta J}{J_{ss}}\right] = \frac{1}{T^2}\mathbb{E}[N]\,\mathbb{V}\left[\frac{\Delta J_i}{J_{i\,ss}}\right] \qquad \Longrightarrow \qquad \mathbb{V}\left[\frac{\Delta J}{J_{ss}}\right] = \frac{(4-\pi)\tau}{\pi T}. \qquad (16)$$

To summarize, all random variables and their corresponding probability distribution function, expectation (mean), and standard deviation (square root of variance) are listed in *Table 1*.

## Numerical simulations

In addition to these analytical calculations of the expectation and variance of the incremental and cumulative uptake, that is, $\Delta J_i/J_{i\,ss}$ over $T_i$ and $\Delta J/J_{ss}$ over $T = \sum T_i$, respectively, we also conducted numerical simulations of contraction events and inter-contraction times. We used a common algorithm that exploits the fact that inter-contraction times are exponentially distributed, and we also simulated inter-contraction times until the maximum time period $T$ is achieved. We checked the histograms, expectation, and variance of both contraction events and inter-contraction times obtained from the numerical simulations and verified that they are all in line with theory, with nearly machine precision error in the expectation values obtained from simulations and theory for the range of $\lambda$ we examined.

For each value of $\lambda$, we conducted 10,000 experiments over a time interval $T$ = 48 hr, and for each experiment, we computed $\Delta J_i/J_{i\,ss}$ and $\Delta J/J_{ss}$. We then calculated the expectation (mean) and variance (square of the standard deviation) of $\Delta J/J_{ss}$ over all 10,000 experiments. Results are shown in the text as a function of $\lambda$ and are in agreement with the theoretical predications in *Table 1*.

## Acknowledgements

Research in the laboratory of TCGB was supported in part by grants from the Deutsche Forschungsgemeinschaft (DFG), the CRC 1182 'Origin and Function of Metaorganisms' (to TCGB) and the CRC 1461 'Neurotronics: Bio-Inspired Information Pathways' (Project-ID 434434223 – SFB 1461) (to TCGB and AK). AK is supported by a DFG grant KL3475/2-1. We thank the Central Microscopy Facility at the Biology Department of the University of Kiel for excellent technical support. TCGB appreciates support from the Canadian Institute for Advanced Research. JN and EK acknowledge support from the National Institute of Health grant 1R01 HL 15362201-A1 and the National Science Foundation INSPIRE grant 1608744.

# Additional information

### Funding

| Funder | Grant reference number | Author |
|---|---|---|
| Deutsche Forschungsgemeinschaft | CRC 1182 "Origin and Function of Metaorganisms" | Christoph Giez<br>Alexander Klimovich<br>Thomas CG Bosch |
| Deutsche Forschungsgemeinschaft | Project-ID 434434223 CRC 1461 "Neurotronics: Bio-Inspired Information Pathways" | Thomas CG Bosch<br>Alexander Klimovich |
| National Institutes of Health | grant 1 R01 HL 15362201-A | Janna C Nawroth<br>Christoph Giez<br>Alexander Klimovich<br>Eva Kanso |
| National Science Foundation | INSPIRE grant 1608744 | Janna C Nawroth<br>Christoph Giez<br>Alexander Klimovich<br>Eva Kanso |

| Funder | Grant reference number | Author |
|---|---|---|
| National Science Foundation | RAISE grant | Eva Kanso |
| Deutsche Forschungsgemeinschaft | KL3475/2-1 | Alexander Klimovich |

The funders had no role in study design, data collection and interpretation, or the decision to submit the work for publication.

## Author contributions

Janna C Nawroth, Conceptualization, Data curation, Formal analysis, Validation, Investigation, Visualization, Methodology, Writing – original draft, Writing – review and editing; Christoph Giez, Data curation, Investigation, Visualization, Methodology, Writing – original draft, Writing – review and editing; Alexander Klimovich, Data curation, Formal analysis, Validation, Investigation, Visualization, Methodology, Writing – original draft, Writing – review and editing; Eva Kanso, Conceptualization, Formal analysis, Supervision, Funding acquisition, Validation, Investigation, Visualization, Methodology, Writing – original draft, Writing – review and editing; Thomas CG Bosch, Conceptualization, Formal analysis, Supervision, Funding acquisition, Investigation, Writing – original draft, Project administration, Writing – review and editing

## Author ORCIDs

Janna C Nawroth (ORCID) http://orcid.org/0000-0003-1898-3968
Christoph Giez (ORCID) http://orcid.org/0000-0002-8101-6498
Alexander Klimovich (ORCID) http://orcid.org/0000-0003-1764-0613
Eva Kanso (ORCID) http://orcid.org/0000-0003-0336-585X
Thomas CG Bosch (ORCID) http://orcid.org/0000-0002-9488-5545

## Decision letter and Author response

Decision letter https://doi.org/10.7554/eLife.83637.sa1
Author response https://doi.org/10.7554/eLife.83637.sa2

# Additional files

## Supplementary files

• MDAR checklist

## Data availability

All data presented in the main manuscript, figures and figure supplements is publicly available at NCBI Bioproject PRJNA842888. The 16S rRNA sequencing raw data are deposited at the SRA and are available under the project ID PRJNA842888.

The following dataset was generated:

| Author(s) | Year | Dataset title | Dataset URL | Database and Identifier |
|---|---|---|---|---|
| Giez C, Klimovich A, Bosch TCG | 2022 | Spontaneous body wall contractions shape and stabilize the symbiotic microbiota | http://www.ncbi.nlm.nih.gov/bioproject/?term=PRJNA842888 | NCBI BioProject, PRJNA842888 |

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
