## [Editor Report]

This important work studies the spontaneous contractions (SC) of the *Hydra* body wall and presents a mathematical model of nutrient transport to hypothesize the role of SC on maintaining the microbiota. The solid evidence presented yields insights into the functional implications of the SC and the increased nutrient update obtained from mixing the local fluid environment through body wall contractions. The main result represents an important observation about the role of hydrodynamics on organism behavior and its relation to diffusive chemical transport processes.

---

## [Decision Letter]

**Decision letter after peer review:**

Thank you for submitting your article "Spontaneous body wall contractions stabilize the fluid microenvironment that shapes host-microbe associations" for consideration by *eLife*. Your article has been reviewed by 2 peer reviewers, and the evaluation has been overseen by a Reviewing Editor and Aleksandra Walczak as the Senior Editor. The reviewers have opted to remain anonymous. We regret the delay in furnishing this decision letter to you.

Essential revisions:

1) The current introduction and references to related work are quite sparse. Previous studies on marine animals such as coral polyps [Kremien et al., Benefit of pulsation in soft corals, PNAS (2013)] have also shown how periodic body contractions may play a role in mixing their local fluid environment to facilitate nutrient transport. The periodic contractions of Yeti crab chelipeds are mentioned, but this only appears in passing at the very end of the Discussion. The authors should expand the introduction/discussion to provide further background on what is known about mixing in other marine organisms. How do the SC of *Hydra* compare regarding timescales and the relevant dimensionless groups (e.g. Re, Pe)?

2) The predicted formula for the uptake is in terms of the characteristic timescale τ of nutrient diffusion. How much variation does τ exhibit for the compounds required by the microbiota, and which microbes consume the compounds corresponding to the greatest value of τ? In principle, it seems these microbes would be the most sensitive to changes in the SC. Is this consistent with what is observed in your experiments?

3) The authors suggest that the exponential distribution of mean inter-contraction (IC) intervals implies an underlying Poisson stochastic process. But there are a number of ways for an exponential distribution to arise, the authors should soften this statement. (It seems perfectly reasonable to use a Poisson process in the model, however.) On the same point, is there any evidence that the authors can point to, perhaps in other organisms, that indicate that a Poisson process is in fact guiding the behavior?

4) The hydrodynamic model is rather simplistic, the body geometry is essentially neglected, so we are really just looking at scaling laws. But it would seem that a more relevant simple calculation would involve, for instance, cylindrical surfaces. The lengths of the *Hydra* and its tentacles may play a much larger role than is currently given credit – for instance the difference in the nutrient uptake might be substantial, based on the logarithmic vs. 1/r scaling of the concentration in 2D vs 3D. Can the authors better justify the choice of a sphere as the most reasonable simple geometrical representation of *Hydra*, even for extracting scaling behaviors?

---

## [Author Response]

Essential revisions:1) The current introduction and references to related work are quite sparse. Previous studies on marine animals such as coral polyps [Kremien et al., Benefit of pulsation in soft corals, PNAS (2013)] have also shown how periodic body contractions may play a role in mixing their local fluid environment to facilitate nutrient transport. The periodic contractions of Yeti crab chelipeds are mentioned, but this only appears in passing at the very end of the Discussion. The authors should expand the introduction/discussion to provide further background on what is known about mixing in other marine organisms. How do the SC of *Hydra* compare regarding timescales and the relevant dimensionless groups (e.g. Re, Pe)?

We thank the reviewer for their helpful suggestions to improve the introduction. We have moderately expanded the introduction and discussion to highlight and contrast known functions of contractions in other marine animals. We do not go into a large discussion of mixing in the introduction because it is not clear upfront that *Hydra* are using spontaneous contractions for fluid mixing, and because not much is known about the role of spontaneous muscle contractions in fluid mixing in other organisms. We thank the referee for pointing out the example of the soft corals. The benefits reported in Kremien et al. do not apply directly to *Hydra* because of differences in length and time scales and flow regimes. Soft corals form large and dense aggregates that greatly exceed the dimensions of *Hydra*, contract rhythmically at 100x higher frequencies than *Hydra* (i.e., not “spontaneously” at irregular intervals), are often exposed to background flows, and operate at intermediate Reynolds numbers that allow for collective bulk flow and fluid advection. We have clarified this important difference and compared scales and dimensionless groups in introduction and discussion.

Mixing by mechanisms other than spontaneous contractions, such as by cilia in filter-feeders or as a “by-product” from swimming, are more prevalent in the literature. But to our knowledge, no other studies have linked spontaneous muscle contractions to a mixing function; indeed, there are very few studies overall on muscle contractions predominantly serving fluid mixing, apart from the rhymically contracting intestinal systems and the soft corals that we now discuss. A comparative study of timescales, Pe, and Re of *Hydra* to other organisms where repetitive muscle contractions may have a dedicated role in fluid mixing is a great idea for future research, but beyond the scope of the present manuscript since little is to be found in literature. The present manuscript aims to first establish this unknown function of spontaneous contractions, using *Hydra* as a model.

2) The predicted formula for the uptake is in terms of the characteristic timescale τ of nutrient diffusion. How much variation does τ exhibit for the compounds required by the microbiota, and which microbes consume the compounds corresponding to the greatest value of τ? In principle, it seems these microbes would be the most sensitive to changes in the SC. Is this consistent with what is observed in your experiments?

The reviewers make an interesting point that would be exciting to explore once more is known about the compounds relevant to *Hydra*’s microbiota. Unfortunately, currently close to nothing is known about the compounds consumed by microbiota that need to be restored, or the compounds generated by the microbiota that need to be removed. Based on the limited knowledge available from other host-symbiont systems such as the mammalian gut, we believe that small molecules, such as amino acids, may play important roles in *Hydra* as well; such molecules would have at least 10 times smaller diffusion coefficients than oxygen, leading to 10 times longer diffusion times for the same distance. We did not speculate on this in the manuscript, however, because of the lack of evidence in literature.

3) The authors suggest that the exponential distribution of mean inter-contraction (IC) intervals implies an underlying Poisson stochastic process. But there are a number of ways for an exponential distribution to arise, the authors should soften this statement. (It seems perfectly reasonable to use a Poisson process in the model, however.) On the same point, is there any evidence that the authors can point to, perhaps in other organisms, that indicate that a Poisson process is in fact guiding the behavior?

We thank the reviewers for the helpful feedback. We agree that exponential distributions in general can arise from many processes; however, an exponential distribution of interval durations between two subsequent events specifically is mathematically equivalent to the presence of a Poisson distribution of the event times (assuming the events are independent; i.e., one event does not trigger the next). Of course, since we are basing this conclusion on the best fit to the experimental interval data, the Poisson distribution is still an estimate but remains the simplest representation of the observed trends. We have clarified this in the text by updating the sentence to “Specifically, the temporal sequence of *Hydra*’s spontaneous contractions can be estimated mathematically by a Poisson distribution of mean λ, given that the distribution of inter-contraction intervals are best fitted by an exponential distribution of mean 1/λ.”

4) The hydrodynamic model is rather simplistic, the body geometry is essentially neglected, so we are really just looking at scaling laws. But it would seem that a more relevant simple calculation would involve, for instance, cylindrical surfaces. The lengths of the *Hydra* and its tentacles may play a much larger role than is currently given credit – for instance the difference in the nutrient uptake might be substantial, based on the logarithmic vs. 1/r scaling of the concentration in 2D vs 3D. Can the authors better justify the choice of a sphere as the most reasonable simple geometrical representation of *Hydra*, even for extracting scaling behaviors?

We thank the reviewer for bringing up this important point.

From a scaling argument, the logarithmic versus 1/r scaling of concentration in 2D versus 3D are properties of the dimensionality of the space, rather than the details of the body geometry. In particular, both a cylinder and a sphere exhibit the same 1/r decay property, as evident in the figures below based on our numerical computations of the concentration field around a cylindrical body of radius a = 1 and length L = 2, compared to the concentration field around a spherical body of radius a = 1.The unsteady diffusion equation outside a sphere has a known analytical solution, while, to our best knowledge, an analytical solution outside of a finite length cylinder is not available and, thus, the concentration field would need to be computed numerically, as we did in the figures below. This presents an additional complication in linking the physics-based model of concentration diffusion to the probabilistic model of contraction events, which we proposed in this manuscript.Importantly, it is not clear that a cylinder is a better model of the *Hydra* geometry. The *Hydra* head has many tentacles that expand in different directions, making it span a spherical domain in the 3D space. Therefore, short of modeling the *Hydra* geometry with its intricate details, the spherical model is as good of an approximation, if not better than the cylindrical geometry.

**Author response image 1. sa2fig1:** Leftmost panels, Numerical solution (and zoom in) of the steady state concentration field around a cylindrical body of length L = 2 and radius r = 1. Middle panels: concentration field around a spherical body of radius 1. Rightmost panel, difference in concentration outside the cylinder and sphere. The difference away from the body surface is less than 5%.